**Data Availability Statement:** All relevant data are within the manuscript and its Supporting Information files.

# Dialysis capacity and nutrition care across Bangladesh: A situational assessment

**Md. Sajjadul Haque Ripon[1], Shakil Ahmed[1], Tanjina Rahman[2]\*, Harun-Ur Rashid[3], Tilakavati Karupaiah[4], Pramod Khosla[5], Zulfitri Azuan Mat Daud[6], Shakib Uz Zaman Arefin[3], Abdus Salam Osmani[7]**

**1** Department of Food Technology and Nutrition Science, Noakhali Science and Technology University, Sonapur, Noakhali, Bangladesh, **2** Institute of Nutrition and Food Science, University of Dhaka, Dhaka, Bangladesh, **3** Kidney Foundation Hospital and Research Institute, Dhaka, Bangladesh, **4** School of Biosciences, Taylor's University, Subang Jaya, Selangor, Malaysia, **5** Department of Nutrition and Food Science, Wayne State University, Detroit, MI, United States of America, **6** Faculty Medicine and Health Sciences, Department of Dietetics, Universiti Putra Malaysia, Serdang, Selangor, Malaysia, **7** National Institute of Kidney Diseases and Urology, Sher-e-Bangla Nagar, Dhaka, Bangladesh

\* tanjina.infs@du.ac.bd

## Abstract

Hemodialysis (HD) is a treatment for ensuring the survival of end-stage kidney disease (ESKD) patients, and nutrition care is integral to their management. We sent questionnaires to evaluate the total dialysis service capacity and nutrition services across all dialysis facilities (DF) in Bangladesh, with responses from 149 out of 166 active DFs. Survey results revealed that 49.7% of DFs operated two shifts, and 42.3% operated three shifts daily, with 74.5% holding between one and ten dialysis machines. Sixty-three percent of DFs served between one and 25 patients per week, and 77% of patients received twice-weekly dialysis. The average cost for first-time dialysis was 2800 BDT per session (range: 2500–3000 BDT), but it was lower if reused dialyzers were used (2100 BDT, range: 1700–2800 BDT). Nutritionists were available in only 21% of the DFs. Parameters related to nutritional health screening (serum albumin, BMI, MIS-malnutrition inflammation assessment, and dietary intakes) were carried out in 37.6%, 23.5%, 2%, and 2% of the DFs, respectively, only if recommended by physicians. Nutrition education, if recommended, was provided in 68.5% of DFs, but only in 17.6% of them were these delivered by nutritionists. The recommendation for using renal-specific oral nutrition supplements (ONS) is not a familiar practice in Bangladeshi DFs and, therefore, was scarcely recommended. Dialysis capacity across Bangladesh is inadequate to meet current or projected needs and nutrition education and support across the DFs to benefit improving patients' quality of life is also inadequate.

## Introduction

Chronic Kidney Disease (CKD) is a significant public health issue globally, with an estimated prevalence of ~13% in 2019 [1]. According to the Global Burden of Disease study, in Bangladesh, CKD affects 11.2 million of people, with an age-standardized rate of 8, 300 cases per

**Funding:** The author(s) received no specific funding for this work.

**Competing interests:** The authors have declared that no competing interests exist.

**Abbreviations:** DF, Dialysis facilities; BMI, Body mass index; SGA, Subjective Global Assessment; nPCR, normalized protein catabolic rate; eGFR, estimated Glomerular Filtration Rate; NGO, Non-governmental organization; ESKD, End Stage Kidney Diseases; ONS, Oral nutrition supplement; pmp, Per Million Population and; CMS.gov, Centers for Medicare & Medicaid Services.

100,000 population. Additionally, there were 16,783 deaths related to CKD, with an age-adjusted rate of 15.4 deaths per100,000population [2]. In 2018, kidney disease was the 9th leading cause of death in Bangladesh, according to the World Health Organization [3]. However, a meta-analysis revealed a higher prevalence of chronic kidney disease (CKD) among Bangladeshi individuals (22.48%) compared to the global prevalence, with females exhibiting a higher prevalence (25.32%) than males (20.31%) [4]. Rashid et al. estimated that there are 20 to 30 million Bangladeshi suffering from CKD, with an ESKD prevalence of 100–120 per million of the population [5–7]. Collectively, these data suggest that some 26 to 30 million Bangladeshi may have CKD, with approximately 2.9 to 3.2 million having CKD stages III to IV.

Individuals with ESKD require kidney replacement therapy (KRT), such as hemodialysis (HD), peritoneal dialysis (PD) or a kidney transplant (KT), to survive [8]. Globally, Around 78% of individuals who require KRT choose dialysis, with almost 89% of them receiving HD [8–10]. In Bangladesh, 33,000 to 41,000 patients may be candidates for KRT, with 50,000 new C KD patients added to this population annually [11,12]. The actual number of patients receiving HD is unknown, and no reliable data is available [13]. Dialysis is a life-saving treatment, but it is also expensive and requires specialized settings, which means its availability is closely linked to resources [9].

Aside from improving dialysis adequacy to improve survival rates, the provision of nutrition education and services for renal patients is a crucial aspect of optimal healthcare. Nutrition education and services can help slow or delay the progression of CKD and aid in the management of HD patients. HD patients are susceptible to protein energy wasting (PEW), which can result in increased morbidity and mortality [14]. In a single dialysis facility in Dhaka, PEW was estimated to be between 17% and 18% [15,16], while a recent study demonstrated that malnutrition is a common problem in Bangladeshi patients undergoing HD [17]. In addition to PEW, fluid and electrolytes imbalances, mineral bone disorders, anemia due to dialysis, uremia-induced metabolic disruptions compound problems for HD patients [18]. Nutrition counseling aimed at moderating sodium, potassium, and phosphorus intake while promoting adequate protein intake is essential for HD patients to improve their outcomes [19]. Rahman et al. found that providing a nutrition education booklet to a Bangladeshi dialysis facility helped renal patients to reduce their serum phosphorus, potassium, and control adequate protein levels and improved their dietary practices [20]. However, successful nutrition management requires careful planning, periodic assessment of nutritional status, as well as monitoring of dietary compliance [19]. As Bangladesh has no formal program to train dietitians, it is unclear how much kidney nutrition care is available to dialysis patients. The current study was therefore conducted to a) evaluate firstly the existing capacity for HD across Bangladesh and b) to assess the extent of kidney nutrition care, if any, that was available at these dialysis facilities. As there is no comprehensive published national renal registry data in Bangladesh to the best of our knowledge, a cross-sectional study was necessary to achieve this first objective.

## Materials and methods

### Study design

A cross-sectional study was carried out in three phases from October 2020 to March 2022 to evaluate the existing capacity of available centers for conducting HD and the extent of nutrition care practices in these centers throughout Bangladesh. In Phase I, we focused on listing all dialysis facilities through Bangladesh. In Phase II, a modified 31-item Questionnaire was developed to face and content validation, and in Phase III, the validated questionnaire was carried out through 64 districts of 8 divisions in Bangladesh.

## Phase I: Listing dialysis facilities

As there is no renal registry in Bangladesh, an initial effort was made to compile a comprehensive list of dialysis facilities across the country using online research and, when necessary, in-person visits. It was observed that there are significant differences in the management of dialysis services across different dialysis facilities in terms of cost, service, nutrition care, and provision of food. Government-owned dialysis facilities typically prioritize accessibility and affordability of services, aiming to provide equitable healthcare to the population. In addition, private facilities prioritize profitability, while non-governmental organizations (NGOs) focus on social or humanitarian missions. To account for these differences, the identified facilities were divided into three sectors; government, private, and NGOs. Overall, a total of 187 dialysis centers were identified, and their contact information, including email addresses, phone numbers, and locations, was recorded for future reference.

## Phase II: Questionnaire development

A modified 31-item Questionnaire (**S1 File**) was used as the data collection tool to collect necessary information required for conducting the current study. Before data collection, the questionnaire was developed based on a previously validated 17-item instrument used in Malaysia following inputs and suggestions from local nephrologists and health care professionals dealing with CKD patients to get access to the maximum amount of information relevant to the scope of this study [21]. The newly developed questionnaire consisted of five sections:

**A:** Characteristics of the dialysis facilities such as sector, charge, times of dialyzer reuse, shift, number of machines and machine operators, number of patients and presence of a nutritionist.

**B:** Nutrition parameters screening of dialysis patients and nutrition education service, providers and materials.

**C:** Recommendation, indications, contraindications, and provision of renal specific oral nutrition supplements (ONS).

**D:** Practice of eating, provision of in-center meal during dialysis and items name of meal.
**E:** Miscellaneous

*Content validity of the questionnaire.* This was carried out with 8 "expert specialists", encompassing nutritionists ($n = 2$) and nephrologists ($n = 6$), as described. Content validation was conducted through a face-to-face interview with each participant. For this method, an expert panel meeting was organized, with the researcher facilitating the content validation process. The participants were requested to provide their feedback using a five-point Likert scale (1 = very poor, 2 = poor, 3 = fair, 4 = good, and 5 = very good) on standard aspects, including i) characteristics of the dialysis facilities, ii) Screening of nutrition parameters, iii) information about oral nutrition supplements (ONS), iv) practice of eating and provision of in-center meal, and v) miscellaneous. The given scores were used to calculate the value for the Content Validity Index (CVI), [22,23] and CVI score was 0.87.

## Phase III: Survey process, sampling and response rate

After identifying 187 dialysis facilities in Bangladesh and developing a questionnaire, the Principal Investigator (PI) and Research Associates (RAs) contacted each facility using available contact information. Of the 166 active facilities, 149 agreed to participate in the study, with 17 not responding. Data were collected from 99 centers through various means including telephone, email, and face-to-face interviews conducted outside of dialysis centers due to the Covid-19 pandemic restrictions. However, data from the remaining 50 centers were collected via physical interview, visiting respective centers and recruited individuals directly involved

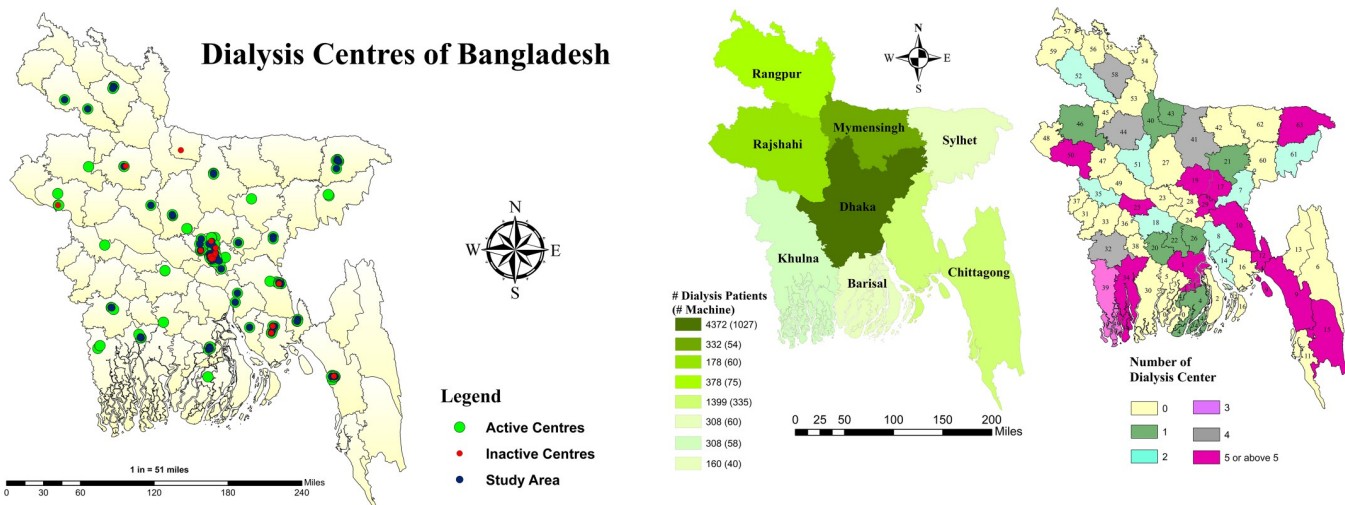

**Fig 1. Survey area. (A)** A total of 149 out of 166 active DFs were selected through simple random sampling from different districts in Bangladesh. **(B)**. Map of Bangladesh showing 8 divisions (left) with number of dialysis patients, machines and 64 districts (right) with number of active DFs.

with dialysis patients in dialysis facilities such as doctors, nurses, nutritionists and dialysis assistants. In this way, the study covered 149 out of 166 dialysis facilities (pluralistic in nature public/private/NGOs) in different districts of the 8 administrative divisions of Bangladesh using convenient sampling, yielding an overall response rate of 90% (**Fig 1**). The 2011 decennial national census was used to report population data for each administrative division and the 64 districts.

During the survey, respondents were asked to provide general comments and recommendations in Bengali at the end of the questionnaire regarding the overall topic of the study for the future development of dialysis facilities in Bangladesh [24]. The responses were documented either manually or recorded and then transcribed as verbatim quotes for qualitative data. A total of 43 statements were categorized thematically. However, in case of respondents who did not agree to record the interview, field notes were collected during the interview.

## Ethics approval and consent to participate

The study protocol received ethical approval from the Research Cell Committee of Noakhali Science and Technology University (NSTU/RC-FTNS/MS/21/73). Informed consent was obtained from both the Institute Review Board of each dialysis facility and individual respondents. Furthermore, a letter of introduction from the Department of Food Technology and Nutrition Science (FTNS) was provided to all participating dialysis facilities before data collection commenced. Additional information regarding the ethical, cultural, and scientific considerations specific to inclusivity in global research is included in the (**S1 Checklist**)

## Statistical analyses

For the mapping of the study area, ESRI ArcGIS version 10.8 was utilized. Shape file format of the maps was obtained from the DIVA-GIS database (https://www.diva-gis.org/), where shape file are provided for free use. The statistical analyses were computed using IBM SPSS version 26 and the statistical significance level was set at $p < 0.05$. For the quantitative analysis, continuous variables that had non-normal distribution were presented as median with interquartile range (IQR). Categorical variables were presented as frequency with percentage. Chi-square

($\chi^2$) test was used to identify associations between categorical variables. Pearson's Chi Square test and Fisher's Exact Chi Square test (when more than 20% cells have expected value less than 5) were used to assess relationships between two categorical variables. Bonferroni post hoc test was used for paired comparisons between groups. Kruskal-Wallis test (k-samples) examined the significance of non-normal distributions of continuous variables with Dunn's comparison used for *post-hoc* analysis.

For qualitative analysis, interviewees were anonymized with random numbers to ensure confidentiality. MAXQDA 2020 qualitative data analysis software was used to manage the written transcripts. The transcriptions were prepared by the PI and checked multiple times to prevent loss of valuable information. Open coding was formulated to understand the essence of the interview responses (commonalities) and then preliminary codes (identifying key phrases) were generated. All the transcripts, field notes, and records were considerably evaluated and addressed to avoid disparities. Finally, after finalizing the codes and categories, inductive themes were generated by applying Braun and Clarke's thematic analysis approach [25].

## Results

### Distribution of DFs

By division, the study identified a total of 187 dialysis facilities (DFs), out of which 166 (89%) were operational. The data showed that 69% of the centers were privately managed, 16% were managed by NGOs, and 15% were managed by the Government (**S1 Table**). The data was further analyzed by district, which highlighted significant disparities in dialysis access throughout the country (**as shown in S2 Table**). The data revealed that out of the 64 districts, 50% (32 districts) did not have any DFs, covering 44.4% of the area and 31.2% of the population. Seven districts had only one DF, while eight districts had two DFs. However, one district, which housed the capital city, covering only 1% of the area and 8.3% of the population, had 50 DFs (30.6% of all DFs).

**Table 1** summarizes the sector-wise distribution of DFs with center characteristics like dialysis shifts, cost, machines, patients and number of dialyzer re-use patterns. About 92% of all DFs run either two or three shifts a day, with a total of 1700 machines in 149 active DFs. Each machine can perform up to 28 dialysis sessions per week, serving a maximum of 14 patients if they go for 2 sessions per week. This indicates that the 1700 machines can serve around 23,800 patients. As per the Kidney Foundation Hospital and Research Institute in Bangladesh, 77% of patients receive dialysis twice weekly, 22% receive thrice weekly, and 1% receive once or four times a week [26] which means approximately 22,300 to 23,800 patients had access to dialysis services in 149 active DFs. In addition, the number of dialysis machines per government-owned center [20 (10–32)] was significantly higher than that of privately-owned centers [6 (4–9)]. Only 6.5% of private centers had more than 20 machines, compared to 43.7% of government centers ($\chi^2$ = 24.7, $p$ <0.001). Nurses (85.2%), medical technicians (42.3%), and medical assistants (20.8%) were responsible for operating the dialysis machines. The percentage of medical technicians operating the machines varied significantly by sector ($\chi^2$ = 6.1, $p$ = 0.048): 64% in the NGO sector, 43.8% in the government sector, and 37% in the private sector.

A dialyzer (filter) is used to purify body fluid and toxic wastes from a patient's blood during hemodialysis. Reuse of a dialyzer means to use it more than once for the same patient. Before reusing a dialyzer, it must be washed and sterilized [27]. Dialyzer reuse trends were different among sector distributions of dialysis facilities ($\chi^2$ = 67.7, $p$ < 0.001). Government centers (93.7%) reused dialyzers more than private (15.7%) or NGO (16.0%) centers. A total of 103 DFs (69.2%) reused a dialyzer several times, while only 46 DFs (30.9%) used a dialyzer for one time, with private centers (41.7%) being more common (refer to Table 1 for reuse of dialyzer pattern).

**Table 1. Sector wise distribution and center's administration of DFs.**

| Parameters in each center | All (n = 149) | Sector | | | p-value |
|---|---|---|---|---|---|
| | | Government (n = 16) | Private (n = 108) | NGO (n = 25) | |
| **# of dialysis shift** | | | | | *Ns* |
| One Shift | 6(4.0) | 3(18.7) | 3(2.8) | 0(0.0) | |
| Two Shifts | 74(49.7) | 7(43.7) | 51(47.2) | 16(64.0) | |
| Three Shifts | 63(42.3) | 5(31.3) | 50(46.3) | 8(32.0) | |
| Four Shifts | 6(4.0) | 1(6.3) | 4(3.7) | 1(4.0) | |
| **# of dialysis machines** | 7 (5–11) | 20 (10–32) | 6 (4–9) | 8 (7–11) | *<0.001ᵃ* |
| **Category (# of Dialysis Machine)** | | | | | *<0.001ᶜ* |
| Below 5 | 38(25.5) | 1(6.3) | 35(32.4) | 2(8.0) | |
| 5 to 10 | 73(49.0) | 4(25) | 52(48.2) | 17(68.0) | |
| 11 to 20 | 21(14.1) | 4(25) | 14(12.9) | 3(12.0) | |
| Above 20 | 17(11.4) | 7(43.7) | 7(6.5) | 3(12.0) | |
| **Dialysis machine operator** | | | | | |
| Nurses | 127(85.2) | 14(87.5) | 91(84.3) | 22(88.0) | *Ns* |
| Medical Technicians | 63(42.3) | 7(43.8) | 40(37.0) | 16(64.0) | *0.048* |
| Medical Assistants | 31(20.8) | 2(12.5) | 24(22.2) | 5(20.0) | *Ns* |
| **Dialysis cost using new dialyzer** | 2800 (2500–3000) | 450 (416–707) | 3000 (2700–3200) | 2500 (2400–2700) | *<0.001ᵇ* |
| **Cost category (new dialyzer)** | | | | | *<0.001ᶠ* |
| Below 1000 BDT | 14(9.4) | 14(87.5) | 0(0.0) | 0(0.0) | |
| 1000–2000 BDT | 5(3.4) | 2(12.5) | 0(0.0) | 3(12.0) | |
| 2001–2500 BDT | 26(17.4) | 0(0.0) | 16(14.8) | 10(40.0) | |
| 2501–3000 BDT | 73(49.0) | 0(0.0) | 61(56.5) | 12(48.0) | |
| Above 3000 BDT | 31(20.8) | 0(0.0) | 31(28.7) | 0(0.0) | |
| **Dialysis cost with dialyzer reuse** | 2100 (1700–2800) | 450 (416–500) | 2300 (2000–3000) | 1900 (1700–1900) | *<0.001ᵇ* |
| **Cost category (Reuse)** | | | | | *<0.001ᵍ* |
| Below 1000 BDT | 16(10.7) | 16(100) | 0(0.0) | 0(0.0) | |
| 1000–2000 BDT | 54(36.2) | 0(0.0) | 31(28.7) | 23(92) | |
| 2001–2500 BDT | 30(20.1) | 0(0.0) | 29(26.9) | 1(4.0) | |
| 2501–3000 BDT | 31(20.8) | 0(0.0) | 30(27.8) | 1(4.0) | |
| Above 3000 BDT | 18(12.1) | 0(0.0) | 18(16.7) | 0(0.0) | |
| **# of Dialyzer Reuse** | | | | | *<0.001ᵈ* |
| One time | 46(30.9) | 0(0.0) | 45(41.7) | 1(4.0) | |
| Two times | 25(16.8) | 1(6.3) | 23(21.3) | 1(4.0) | |
| Three times | 42(28.2) | 0(0.0) | 23(21.3) | 19(76.0) | |
| Above Three times | 36(24.2) | 15(93.7) | 17(15.7) | 4(16.0) | |
| **# of patients/center** | 30 (15–56) | 68 (40–165) | 25 (15–47) | 35 (16–55) | *0.001ᵃ* |
| **Category (# of patients)** | | | | | *0.001ᵉ* |
| <25 | 63(42.3) | 2(12.5) | 51(47.2) | 10(40.0) | |
| 25–50 | 46(30.9) | 2(12.5) | 35(32.4) | 9(36.0) | |
| 50–100 | 27(18.1) | 7(43.7) | 17(15.7) | 3(12.0) | |
| >100 | 13(8.7) | 5(31.2) | 5(4.6) | 3(12.0) | |

Data is presented as either n (%) or median with interquartile range (IQR).

Independent-Samples *Kruskal-Wallis* and *Chi square* ($\chi^2$) test was performed and *p <0.05* is considered significant. *Bonferroni post hoc* test indicates significant difference between sectors; ns: Not significant.

[a]*Bonferroni post hoc* test indicates *p < 0.017* for pairwise comparison between private vs. government.

[b]*Bonferroni post hoc* test indicates *p < 0.017* for pairwise comparison between government vs. private and NGO vs. private.

[c]*Bonferroni post hoc* test have been used after chi square indicates *p < 0.004* for pairwise comparison between government vs. private in terms of dialysis machine above 20

[d]in terms of maximum number of use dialyzer above three and

[e]in terms of dialysis patients above 100.

[f] *Bonferroni post hoc* test have been used after chi square indicates *p < 0.003* for pairwise comparison between government vs. private in terms of 1st time dialysis charge below 1000 BDT and in terms of reusing dialyzer dialysis charge below 1000 BDT.

The cost of dialysis with a new dialyzer in private centers differed significantly from government centers [3000 (2700–3200) vs. 450 (416–707)] and NGO centers [3000 (2700–3200) vs. 2500 (2400–2700) BDT respectively]. The government centers had more DFs with a new dialyzer cost below 1000 BDT compared to private or NGO centers ($\chi^2$ = 106.3, $p$ < 0.001). Similarly, the cost of dialysis with a reused dialyzer in private centers differed significantly from government [2300 (2000–3000) vs. 450 (416–500)] and NGO centers [2300 (2000–3000) vs. 1900 (1700–1900)] BDT respectively. The government centers had more DFs with a reused dialyzer cost below 1000 BDT compared to private or NGO centers ($\chi^2$ = 117.577, $p$ < 0.001).

The number of patients per center varied significantly between government centers [68 (40–165)] and private centers [25 (15–47)]. The sector distribution of dialysis facilities significantly correlated with the categories of centers dialyzing more than 100 patients ($\chi^2$ = 21.04, $p$ = 0.001), with government centers having the highest number of centers (n = 5, 31.2%), followed by NGO centers (n = 3, 12.0%) and private centers (n = 5, 4.6%).

## Sector and geographical region wise distribution of DFs with nutritionist accessibility

It appears there is no registered dietitian (RD)as conventionally understood [28] available in Bangladesh. Typically, in hospitals in urban locations nutrition services are provided by "*pushtibid*" or "*nutritionists*", holding a 4-year Bachelor's with or without a one-year of master's degree (by course work) in nutrition and food science.

Table 2 summarized nutritionist accessibility of 149 selected centers. The results revealed that only 18.8% of government institutes provided access to nutritionists when needed, while private institutes had a higher accessibility rate of 24%. In contrast, NGO-related institutes had the lowest accessibility rate at 8.0%.

DFs in *Dhaka* are centrally located compared those in other regions such as *Chittagong* in the South-east, *Barisal* and *Khulna* in the South-west, *Mymensingh* and *Sylhet* in the North-east, *Rajshahi* and *Rangpur* in the North-west regions. Results showed that HD centers in the central region had the highest accessibility rate of nutritionists, with 24.6% of the 69 DFs having access to nutritionists, followed by the south-east region with 10.8% of the 37 DFs having access. The south-west region had 37.5% of the 16 DFs with access, the north-east region had 28.6% of the 14 DFs with access, and the north-west region had no access to nutritionists in

**Table 2. Nutritionist accessibility in different DFs.**

| | All,<br>n (%) | Nutritionist Accessibility | | |
|---|---|---|---|---|
| | | Nutritionist available, n (%) | Nutritionist not available, n (%) | *p*-value |
| **Sectors Wise** | **149 (100)** | **31 (20.8)** | **118 (79.2)** | ***ns*** |
| Government | 16 (10.7) | 3 (18.8) | 13 (81.2) | |
| Private | 108 (72.5) | 26 (24.0) | 82 (76.0) | |
| NGO | 25 (16.8) | 2 (8.0) | 23 (92.0) | |
| **Geographical Regions Wise** | | | | ***0.034*** |
| Central | 69 (46.3) | 17 (24.6) | 52 (75.4) | |
| North East | 14 (9.4) | 4 (28.6) | 10 (71.4) | |
| North West | 13 (8.7) | 0 (0.0) | 13 (100.0) | |
| South West | 16 (10.7) | 6 (37.5) | 10 (62.5) | |
| South East | 37 (24.8) | 4 (10.8) | 33 (89.2) | |

Data were collected from 149 active DFs throughout Bangladesh. *p<0.05* was considered significant using *Chi Sq ($\chi^2$)* test. Data is presented as n (%); ns: Not significant; NGO: Non-governmental organization.

any of the 13 dialysis facilities. Overall, 31 out of 149 (20.8%) DF had access to nutritionists and the remaining 118 active DF (79.2%) had no access to any nutritionist. The difference in nutritionist accessibility between the geographical regions was found to be statistically significant ($\chi^2$ = 9.75, $p < 0.034$).

## Nutrition screening and nutrition education

The study evaluated the monitoring of nutrition parameters and nutrition education in a sample of 149 DF (Table 3). The results revealed that 85 DF (57%) did not conduct any nutrition monitoring for patients. Of the DF that did monitor, 56 DF (37.6%) measured serum albumin, and 35 DF (23.5%) measured BMI at the time of patient admission. Only 3 DF (2%) monitored dietary intake and MIS, but this was only done when a nutritionist was available. However, significant differences were found between nutritionist accessibility and monitoring of BMI ($\chi^2$ = 21.41, $p < 0.001$) and serum albumin ($\chi^2$ = 12.10, $p = 0.001$). Among the 31 DF with nutritionist access, 54.8% screened for BMI, and 64.5% screened for serum albumin. Furthermore, the majority of centers not performing nutrition monitoring (64.4%) reported the unavailability of a nutritionist ($\chi^2$ = 12.54, $p < 0.001$). However, no center used any available nutrition screening tools such as SGA or nPCR.

**Table 3. Nutrition screening and provision of nutrition education in different DFs based on sector and nutritionist accessibility.**

| Parameters for Nutrition Screening and Practice | All, n = 149 | Sector | | | p- value | Nutritionist Accessibility | | p-value |
|---|---|---|---|---|---|---|---|---|
| | | Govt. n = 16 | Private, n = 108 | NGO, n = 25 | | Nutritionist available, n = 31 | Nutritionist not available, n = 118 | |
| **Which nutrition parameter(s) is/are monitored?** | | | | | | | | |
| BMI | 35(23.5) | 6(37.5) | 25(23.1) | 4((16.0) | Ns | 17(54.8) | 18(15.3) | <**0.001** |
| Serum Albumin | 56(37.6) | 8(50.0) | 40(37.0) | 8(32.0) | Ns | 20(64.5) | 36(30.5) | **0.001** |
| Dietary Intake | 3(2.0) | 1(6.2) | 2(1.9) | 0(0.0) | * | 3(9.7) | 0(0.0) | * |
| MIS | 3(2.0) | 0(0.0) | 1(0.9) | 2(8.0) | * | 3(9.7) | 0(0.0) | * |
| Nothing is monitored | 85(57.0) | 6(37.5) | 62(57.4) | 17(68.0) | ns | 9(29.0) | 76(64.4) | <**0.001** |
| **Providing nutrition education?** | | | | | ns | | | <**0.001** |
| Yes | 102(68.5) | 11(68.8) | 73(67.6) | 18(72.0) | | 30 (96.8) | 72 (61.0) | |
| No | 47(31.5) | 5(31.2) | 35(32.4) | 7(28.0) | | 1 (3.2) | 64 (54.2) | |
| **Who delivers nutrition education?** | | | | | | | | |
| Doctors | 94(92.2) | 10(90.9) | 67(91.8) | 17(94.4) | ns | 24(80.0) | 70(59.3) | ns |
| Nutritionists | 18(17.6) | 2(18.2) | 14(19.2) | 2(11.1) | ns | 18(60.0) | 0(0.0) | <**0.001** |
| Nurses | 17(16.7) | 3(27.3) | 11(15.1) | 3(16.7) | ns | 4(13.3) | 13(11.0) | ns |
| Medical Assistants | 16(15.7) | 1(9.1) | 11(15.1) | 4(22.2) | Ns | 6(20.0) | 10(8.5) | ns |
| **How frequently is nutrition education provided?** | | | | | Ns | | | 0.018 |
| Regular Basis | 8(7.8) | 0(0.0) | 6(8.2) | 2(11.1) | | 5(16.7) | 3(2.5) | |
| As Per Required | 93(91.2) | 11(100) | 66(90.4) | 16(88.9) | | 24(80.0) | 79(66.9) | |
| Others | 1(1.0) | 0(0.0) | 1(1.4) | 0(0.0) | | 1(3.3) | 0(0.0) | |
| **How is nutrition education delivered?** | | | | | Ns | | | 0.022 |
| Individuals | 86(84.3) | 9(81.8) | 62(84.9) | 15(83.3) | | 22(73.3) | 64(54.2) | |
| Group Sessions | 3(2.9) | 0(0.0) | 2(2.7) | 1(5.6) | | 0(0.0) | 3(2.5) | |
| Both | 13(12.7) | 2(18.2) | 9(12.3) | 2(11.1) | | 8(26.7) | 5(4.2) | |

Data were collected from 149 active DFs throughout Bangladesh and $p < 0.05$ was considered significant using Chi Sq ($\chi^2$) test. Data is presented as n (%); ns: Not significant.

*Chi Square ($\chi^2$) Test is not applied because of having minimum expected value less than one.

Abbreviation: BMI- body mass index.

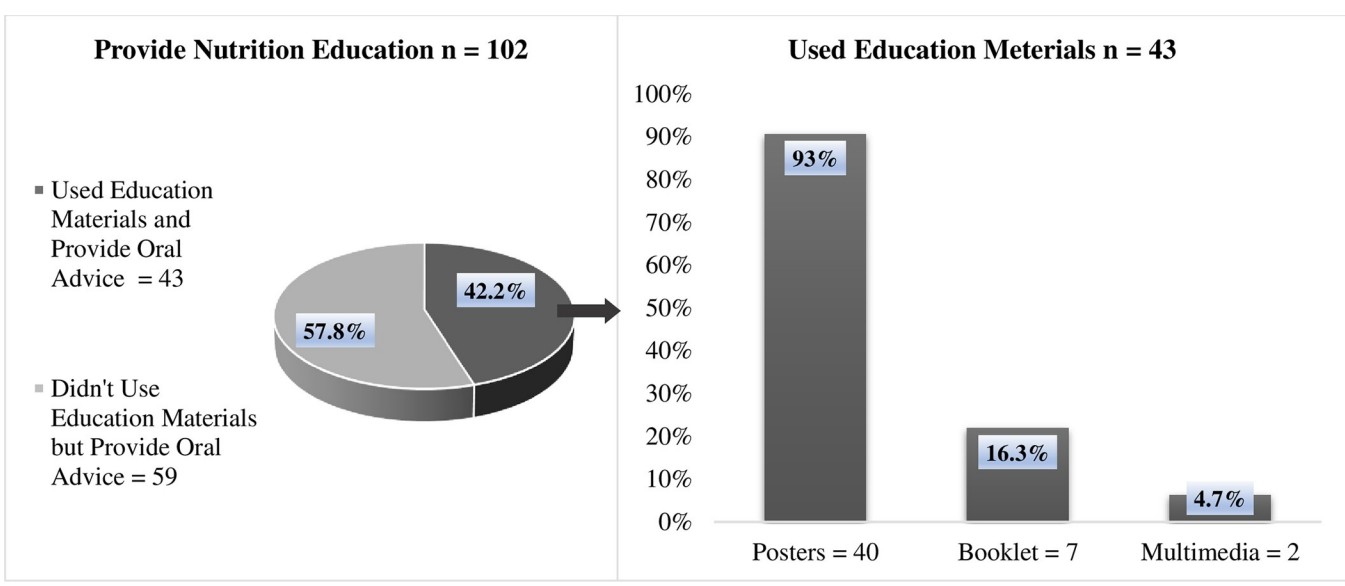

**Fig 2. DFs providing nutrition education.**

There was no significant association found ($p > 0.05$) between the sector distribution of DF and the provision of nutrition education and it was observed that nutrition education was available in 102 DF (68.5%). However, there was a significant difference in nutrition education provision based on nutritionist access ($\chi^2 = 14.54$, $p < 0.001$). Of the 31 DF with nutritionist access, 96.8% provided nutrition education compared to 61% of centers without nutritionist access. Nutrition education was provided by various healthcare professionals, with nutritionists delivering education in 18 out of 31 dialysis facilities (60%) and the correlation between nutritionist accessibility and provision of nutrition education was significant ($\chi^2 = 72.56$, $p < 0.001$). In addition, frequency of nutrition education ($\chi^2 = 6.63$, $p = 0.018$) and manner of nutrition education ($\chi^2 = 7.18$, $p = 0.022$) significantly correlated to nutritionist availability, and mostly delivered as per requirement of patients ($n = 93$, 91.2%). In most cases, nutrition education was delivered to patients individually ($n = 86$, 84.3%) via face-to-face counseling.

In **Fig 2,** the scenario of the provision of nutrition education by DF were recorded in this study, where, 102 DF (68.5%) were reported to provide nutrition education, delivered only orally (57.8%) or combined with education material (42.2%). Education materials were in the form of posters ($n = 40$, 93%), booklet ($n = 7$, 16.3%), and multimedia ($n = 2$, 4.7%).

### Provision of in-center meals and renal specific ONS

**Table 4** showed the availability of in-center meals and the use of oral nutrition supplements (ONS) specifically designed for patients with renal disease. The results indicate that 92.6% (138) of the dialysis facilities (DFs) allowed patients to eat during their dialysis treatment. This provision was significantly associated with the type of facility, as shown by the sector-wise distribution of private (93.5%), NGO (100%), and government (75.0%) facilities ($\chi^2 = 7.23$, $p = 0.017$). Furthermore, 49.7% (74) of the DFs provided meals during dialysis, and among them, 93.2% (69) reported providing snacks such as *singara*, sandwich, *semai*, noodles, and eggs.

Furthermore, we inquired about the availability of complete nutrition supplements specifically designed for people with kidney disease, which are low in potassium, phosphorus, and sodium, and contain a blend of high proteins, carbohydrates, and fats, such as *Nepro*, *Renalcal*,

**Table 4. Provision of in-center meals and ONS practice in DFs based on sector and nutritionist accessibility.**

| Nutrition Practice ONS and in-center Meals | All, n = 149 | Sector | | | p-value | Nutritionist Accessibility | | p- value |
|---|---|---|---|---|---|---|---|---|
| | | Govt. n = 16 | Private, n = 108 | NGO, n = 25 | | Nutritionist available, n = 31 | Nutritionist not available, n = 118 | |
| **Does center allow eating during dialysis?** | | | | | *0.017* | | | *Ns* |
| Yes | 138(92.6) | 12(75.0) | 101(93.5) | 25(100) | | 28(90.3) | 110(93.2) | |
| No | 11(7.4) | 4(25.0) | 7(6.5) | 0(0.0) | | 3(9.7) | 8(6.8) | |
| **Does center provide meals during dialysis?** | | | | | *0.003* | | | *Ns* |
| Yes | 74(49.7) | 2(12.5) | 56(51.9) | 16(64.0) | | 18(58.1) | 56(47.5) | |
| No | 75(50.3) | 14(87.5) | 52(48.1) | 9(36.0) | | 13(41.9) | 62(52.5) | |
| **Types of meal provided** | | | | | *ns* | | | *Ns* |
| Full Meal | 5(6.8) | 0(0.0) | 5(8.9) | 0(0.0) | | 3(16.7) | 2(3.6) | |
| Light Meal | 69(93.2) | 2(100) | 51(91.1) | 16(100) | | 15(83.3) | 54(96.4) | |
| **Recommendation for renal specific (ONS)** | | | | | * | | | * |
| Yes | 0(0.0) | 0(0.0) | 0(0.0) | 0(0.0) | | 0(0.0) | 0(0.0) | |
| No | 149(100) | 16(100) | 108(100) | 25(100) | | 31(100) | 118(100) | |
| **Reason for not suggesting ONS?** | | | | | | | | |
| Not Available | 59(39.6) | 10(62.5) | 40(37.0) | 9(36.0) | *ns* | 10(32.3) | 49(41.5) | *Ns* |
| Costly | 62(41.6) | 9(56.3) | 42(38.9) | 11(44.0) | *ns* | 11(35.5) | 51(43.2) | *ns* |
| Not indicated by HCP | 87(58.4) | 5(31.3) | 68(63.0) | 14(56.0) | *ns* | 20(64.5) | 67(56.8) | *ns* |

Data were first collected from 149 active DFs throughout Bangladesh and *p-value < 0.05* was considered significant using *Chi Sq (χ2)* test. Data were presented as n (%); ns: Not significant.

Abbreviation: ONS- oral nutrition supplement, HCP-Health Care Professionals.

* No statistics are computed because ONS recommendation value is a constant.

*Prorenal*, *Novasource Renal*, and *Suplena*. However, none of the dialysis facilities (DF) reported recommending any renal-specific oral nutrition supplements (ONS). The reasons for this included a lack of indication by healthcare professionals (n = 87, 58.4%), unavailability of ONS products (n = 59, 39.6%), and the high cost of ONS products (n = 62, 41.6%).

## Stakeholder recommendations

This study also processed "general open comments" from the respondents to improve future dialysis services (**Table 5**). Recommendations that could help in the overall improvement of future dialysis services were categorized into sub-themes which included increasing the

**Table 5. Recommendations for improving future dialysis services.**

| Theme | Sub-Theme | Illustrative quotes |
|---|---|---|
| Nutrition | Improvement of nutrition Care | *Provision of training on medical nutrition therapy for DF staff* |
| | | *Increase awareness about nutrition in patient management among medical professionals.* |
| | Increase incentives for Nutritionist | *Improve the salary structure for nutritionists to attract more graduates from nutrition background* |
| | Improve patients' attitude | *Adopt patient-centered approaches to enhance patients' attitudes towards nutrition care.* |
| Healthcare | Improvement of medical services | *Increase the number of specialized doctors and staff.* |
| | | *Ensure the availability of necessary diagnostic facilities, drugs, and medicines for patients.* |
| | Reduce waiting times | *Increase the number of dialysis beds and ensure that all dialysis machines are active.* |
| Health awareness | Increase hygiene Practice | *Prioritize and enforce appropriate hygiene practices to reduce the risk of infections.* |
| Psychological Accommodation | Improve psychological care | *Provide emotional supportive care for patients who face multiple co-morbidities.* |
| | Financing | *Provision of medical insurance or government subsidies for Psychological Relief.* |

number of centers, manpower and equipment in hospitals, dissemination of awareness about dialysis initiation, nutritional and psychosocial care of patients.

## Discussion

### DF characteristics

The objective of this study was to evaluate the current capacity for hemodialysis (HD) and nutrition care and support in dialysis facilities (DFs) across Bangladesh. During the period of 2009–2010, there were 84 dialysis facilities (DFs) present in Bangladesh [7], and the country's total population was reported to be 142 million in 2011 which indicates there is only one DF for 1,694,274 people. In contrast to the USA, there are 7,500 dialysis clinics for a population of 331 million as of April 26, 2021 (CMS.gov United States Census Bureau). Malaysia's National Renal Registry reported 667 dialysis facilities in 2015, providing access to one facility for every 49,026 people with a population of 32,700,000 (Department of Statistics Malaysia). In addition, our study revealed that only 1700 dialysis machines are available in 149 active DFs to cater to 22,300 to 23,800 patients, and out of these, 300 machines are located in just two DFs situated in Dhaka city, namely the Kidney Foundation Hospital and Research Institute and Gonoshasthaya Kendra dialysis center.

The Academy of Nutrition and Dietetics has reported that standards for dialysis management and nutrition care can vary across different regions, districts, and sectors [29]. According to our study, nutritionists are more accessible in private dialysis facilities located in the southwest, central, and northeast regions of the country. Furthermore, KRT practices differ significantly between developing and developed nations, with developing countries having poorer dialysis infrastructure, higher dropout rates, inadequate staffing, greater out-of-pocket expenses, and fewer successful kidney transplant options [30].

Dialysis adequacy is important in reducing complications for patients and is influenced by various factors including co-morbidities of patients, economy, distance to DF, supervision of nephrologists. A study in Dhaka, Bangladesh found that patients receiving 12 hours per week of hemodialysis had better adequacy (43%) compared to those receiving only 8 hours (21%) [31]. According to the KDOQI Guidelines of 2006, incremental HD is suggested to maintain residual renal function at estimated glomerular filtration rate (eGFR) less or equal 15 ml/min/1.73m$^2$ [32], while in Bangladesh, most HD patients starts at eGFR of 6–7 ml/min/1.73m$^2$. To emphasize the point, it should be noted that conserving residual renal function is not a consideration when deciding to initiate dialysis in Bangladesh. Due to the high cost of treatment and limited availability of HD facilities, patients often begin dialysis with only 2 sessions per week. Furthermore, due to the aforementioned factors, patients are often reluctant to undergo 3 sessions per week [33].

According to our study, the cost of hemodialysis (HD) per session in Bangladesh varies depending on the type of facility and these costs are comparable to other Asian countries such as India ($36.85-$49.2 USD or 3000–5000 Rs), Malaysia ($22.52 USD or 100 RM), and Pakistan ($4.44-$22.23 USD or 1000-5000 Rs-Pks) [30]. The reuse of dialyzers is a common practice in developing countries. By reusing dialyzers, the hourly cost of dialysis can be reduced by 33% [34]. Our study found that with the reuse of dialyzers, patients need to spend $27 USD in private facilities and $21 USD in NGO facilities per session of HD, which is comparatively lower than the cost associated with using a new dialyzer. Since the re-use of dialyzers is safe, DF focus on issues of reduced dialyzer efficiency and patient outcomes [35].

A recent global study found that only 74 out of 155 countries have access to dietitians and dietary services for renal-specific nutrition. In low-middle income countries, such services were available in only 35% and never available in 23% of low-income countries [36]. In

Bangladesh, only 20.8% of dialysis facilities had accessibility to nutritionists, with availability influenced by geographical regions and type of dialysis provider. This limited nutrition care scenario is similar to that of Malaysia (~33%) [21] and other Southeast and South Asian countries, where access to dietitians for dialysis patients is also limited compared to European countries [37,38].

## Tasks of the nutrition care process

The delivery of nutrition in dialysis facilities is subject to various factors such as institutional policies, government regulations, professional ethical codes, and environmental trends, even in the United States [39]. Therefore, depending solely on physicians to carry out this task may have a negative impact on the quality of care provided [19]. A study conducted in Australia found that the incidence of malnutrition in dialysis patients decreased significantly from 14% to 3% when nutrition management was carried out by dietitians or when patients followed recommended changes in protein and diet intake [40]. Another point, It is essential to provide renal-specific educational tools that are aligned with international practice guidelines such as KDOQI to ensure optimal control over diet-related issues in dialysis patients [41]. Standardized educational tools like these are crucial for achieving optimal control over diet-related issues, including hypertension, hyper/hyponatremia, hypo/hyperphosphatemia, and hypo/hyperkalemia in dialysis patients [42].

In addition, nutrition screening, being the initial step in the nutrition care process, enables the identification of malnourished dialysis patients for necessary nutritional interventions [43,44]. However, our study revealed that nutrition screening was significantly limited in Bangladesh DFs. Furthermore, regular monitoring of patients' nutritional parameters was not conducted in these facilities. Renal dietitians possess specialized skills to identify malnutrition in patients through validated screening methods such as subjective global assessment (SGA), malnutrition inflammation score (MIS), weight loss estimation, assessing appetite and diet intake [45–47].

To enhance patient care in dialysis facility, it is essential to develop and certify nutritionists with advanced renal dietetic skills, particularly for managing balanced diet in HD patients. Renal dietitians have the advantage of regular monthly follow-ups with dialysis patients, which foster a unique relationship. The Nutrition Care Process (NCP) enables precise nutrition diagnoses that reflect the intricate involvement of renal dietitians with dialysis patients [48]. In Korea, nutritional assessment was uncommon (17%) due to a lack of renal nutrition specialists, particularly in rural areas. However, it is crucial to prioritize and incorporate assessments, such as SGA and MIS in the future to improve dialysis patients' nutritional outcomes [49]. A study has shown that providing an adaptive diet to once-weekly hemodialysis patients can yield positive results. The diet includes low-protein and low-phosphorus options on non-dialysis days, while allowing higher protein intake on dialysis days. This approach has demonstrated potential benefits, including lower β2-microglobulin levels, improved phosphorus control, and reduced doses of erythropoiesis-stimulating agents [50]. Another study in Bangladesh showed that dialysis patients over the age of 40 years with insufficient protein intake were at a higher risk of malnutrition than younger patients with adequate protein intake. Underweight patients (BMI<18.5 kg/m2) were also at a greater risk of developing malnutrition. This study also indicated that Malnutrition was more prevalent in anemic patients (18%) [51].

## Regarding provision of in-center meals and renal-specific ONS

Providing in-center meals and oral nutrition supplements specifically for renal patients during dialysis treatment sessions are highly effective interventions for improving serum albumin

levels, BMI, and overall longevity and quality of life [52,53]. Globally, ONS is available in 81% of 155 countries and recommended to their dialysis patients [36]. On the other hand, recommendation of renal-specific ONS is not a familiar practice in Bangladeshi DF and it is not recommended to patients. According to our study, most DF (92.6%) allowed eating during dialysis. However, eating during dialysis is a controversial practice. In the USA, it is not allowed due to concerns about postprandial hypotension, choking, infection, and diabetes and phosphorus control. In contrast, In most European and Southeast Asian countries, dialysis outpatients are routinely given meals and those who eat during treatments have better nutritional status, serum albumin, and survival rates compared to the USA [54–56].

### Recommendations to improve existing dialysis services

A recent study reported that, a lack of trained personnel especially in nephrology is a critical cause of increasing global burden of CKD. The global nephrologists' density was 8.83 per million populations (PMP) mostly in high-income countries, but as low as 0.65 PMP in low-income or developing countries like Bangladesh [57]. When it comes to nutrition, however, in Bangladesh, there may be fluctuations in nutrition care due to the lack of expertise in nutrition-related parameter monitoring and education. Renal nutrition support personnel (RNSP) could assist in this regard, as doctors may be occupied with patient care and other healthcare professionals may lack the necessary skills. Only those with a minimum of a graduate degree in 'Nutrition and Food Science' or a master's degree in the field are qualified to receive RNSP training and perform the role adequately. It is imperative to organize a national-level training program to train RNSPs promptly and aid advanced-stage renal patients in the country.

### Limitations and strengths

The study's observations were reliant on survey responses from DFs since no published data was accessible. This approach poses a limitation as the gathered information is subjective in nature. For instance, reported information on regular monitoring of serum albumin may be overestimated (by 38% of DFs) as we were unable to verify actual data monitoring. Furthermore, since this study was unfunded, data collection was aided by graduate students specializing in Nutrition and Food Science from a reputable public university in Bangladesh. Moreover, the major strengths of this study was for the first time a situational analysis was carried out on the dialysis capacity and nutrition practice in Bangladeshi dialysis facilities as a nationwide survey which generated both quantitative and qualitative data to reflect on the actual scenario in Bangladesh. Open comments from different stakeholders allowed identifying thematic issues to improve the existing state of dialysis facilities in terms of nutrition, healthcare facilities, health awareness, psychological accommodation.

### Conclusion

Our study evaluated the nutrition care practices and dialysis management information in Bangladeshi dialysis facilities. We found a significant variation in services, where the quality of care depended largely on cost. Unfortunately, the standard of nutrition care was generally inadequate due to the lack of attention given to patient-centered approaches and standardized guidelines. To improve the situation, we recommend the establishment of a national renal registry system to monitor the number of patients on advanced stages of CKD, which can help with trend analysis and guide future strategies. We also suggest a national training program to create a group of renal nutrition support personnel (RNSP) who can provide renal-specific nutrition care and support to advanced CKD patients. These RNSPs can work with healthcare professionals to standardize nutrition practices and improve nutrition care, an essential aspect

of comprehensive treatment for HD patients. We hope that the findings of our study will prompt appropriate authorities to take necessary steps to raise community-level awareness and implement national-level training programs to improve the quality of nutrition care in healthcare centers throughout the country, ultimately improving the quality of life for disadvantaged populations in resource-poor settings like Bangladesh.

## Supporting information

**S1 Checklist. Inclusivity in global research.**
(DOCX)

**S1 Table. Distribution of DFs across Bangladesh based on geographic divisions.** This is the S1 Table legend: [a]Bangladesh is divided into 8 administrative divisions which are further divided into 64 districts. Data for the individual districts can be found in S2 Table. [b]Total number of centers. [c]Number of centers that are Private, NGO or Government operated. An additional 21 centers were not active during the data collection period (14 in Dhaka, 4 in Chittagong, 2 in Rajashahi and 1 in Mymensingh). [d]The sum of the population of each division as reported in the 2011 decennial national census. The 2021 national census has been delayed due to Covid-19. Current total population of Bangladesh as of Feb 2022 is ~ 167 million.
(DOCX)

**S2 Table. Comparison of active number of DFs with population in each district of 8 divisions in Bangladesh.** This is the S2 Table legend: [1]Data were collected from Population & Housing Census (2011) by Bangladesh Bureau of Statistics (BBS).
(DOCX)

**S1 File. The modified 31-item questionnaire.**
(DOCX)

## Acknowledgments

We would like to thank all dialysis managers, doctors, nutritionists from different dialysis facilities. Special thanks to Kidney Foundation Hospital and Research Institute (KFHRI) as they helped us to obtain some recent data. Also thanks to the faculties and students from the department of *Food Technology and Nutrition Science* (FTNS), Noakhali Science and Technology University (NSTU), Bangladesh–Moumita Dey, Syeda Saima Alam, Rafiuzzaman Rafi, Marium Sultana, Farhana Akhtar, Nasrin Sultana, Arafat, Mahmudul Hasan, Pronoy Dey, Meraz Tanvir, Mohammad Sajid Quraishi and Faria Sharmin for their contribution in data collection to completion of the study.

## Author Contributions

**Conceptualization:** Tanjina Rahman, Tilakavati Karupaiah, Pramod Khosla.

**Data curation:** Tanjina Rahman, Harun-Ur Rashid, Tilakavati Karupaiah, Pramod Khosla, Zulfitri Azuan Mat Daud, Shakib Uz Zaman Arefin.

**Formal analysis:** Md. Sajjadul Haque Ripon, Shakil Ahmed, Tanjina Rahman.

**Investigation:** Md. Sajjadul Haque Ripon, Shakil Ahmed, Tanjina Rahman.

**Methodology:** Md. Sajjadul Haque Ripon, Shakil Ahmed, Tanjina Rahman, Zulfitri Azuan Mat Daud.

**Project administration:** Harun-Ur Rashid, Tilakavati Karupaiah, Pramod Khosla.

**Resources:** Harun-Ur Rashid, Abdus Salam Osmani.

**Software:** Md. Sajjadul Haque Ripon, Shakil Ahmed, Tanjina Rahman.

**Supervision:** Tanjina Rahman, Harun-Ur Rashid, Tilakavati Karupaiah, Pramod Khosla, Zulfitri Azuan Mat Daud, Shakib Uz Zaman Arefin.

**Validation:** Harun-Ur Rashid, Shakib Uz Zaman Arefin, Abdus Salam Osmani.

**Visualization:** Shakil Ahmed, Tanjina Rahman, Zulfitri Azuan Mat Daud.

**Writing – original draft:** Md. Sajjadul Haque Ripon, Shakil Ahmed, Tanjina Rahman.

**Writing – review & editing:** Shakil Ahmed, Tanjina Rahman, Harun-Ur Rashid, Tilakavati Karupaiah, Pramod Khosla, Zulfitri Azuan Mat Daud, Shakib Uz Zaman Arefin, Abdus Salam Osmani.

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
