## [Decision Letter · Decision Letter 0]

20 Jun 2023

PONE-D-23-11624Dialysis Capacity and Nutrition Care across Bangladesh: A Situational AssessmentPLOS ONE

Dear Dr. Rahman,

Thank you for submitting your manuscript to PLOS ONE. After careful consideration, we feel that it has merit but does not fully meet PLOS ONE’s publication criteria as it currently stands. Therefore, we invite you to submit a revised version of the manuscript that addresses the points raised during the review process.

We look forward to receiving your revised manuscript.

Kind regards,

Ankur Shah

Academic Editor

PLOS ONE

Additional Editor Comments:

The manuscript is improved from the prior submission but would still benefit from further revision, please see the suggestions of the reviewers.

Reviewers' comments:

Reviewer's Responses to Questions

**Comments to the Author**

1. Is the manuscript technically sound, and do the data support the conclusions?

Reviewer #1: Yes

Reviewer #2: No

Reviewer #3: Yes

2. Has the statistical analysis been performed appropriately and rigorously? 

Reviewer #1: Yes

Reviewer #2: Yes

Reviewer #3: I Don't Know

3. Have the authors made all data underlying the findings in their manuscript fully available?

Reviewer #1: Yes

Reviewer #2: Yes

Reviewer #3: Yes

4. Is the manuscript presented in an intelligible fashion and written in standard English?

Reviewer #1: Yes

Reviewer #2: Yes

Reviewer #3: Yes

5. Review Comments to the Author

Reviewer #1: Thanks to the authors for modifying the manuscript, however, I suggest performing the following based on my previous comments:

Add a paragraph in the methods to illustrate the reasons you divided the dialysis facilities into 3 sectors.

Reviewer #2: This manuscript is a situational assessment of Bangladesh's dialysis capacity and nutrition care.

The report describes the number and the location of dialysis centres in Bangladesh, underlying that the distribution is not homogenous in the country and the number of centres is inferior compared with other countries in the world. The nutrition education of patients throughout the country is deficient and should be improved.

The manuscript is a descriptive epidemiological report and a limited value since there is no description of the biological parameters of the dialytic procedures.

Moreover, the problem of nutrition is superficially described without specific details.

Reviewer #3: The article aims to provide a "situational report" on dialysis capacity and nutrition care in Bangladesh. While the study does address these objectives, there are various errors in the write-up which need to be corrected before publication.

1. Several grammatical errors and unclear sentences. Outlined are a few:

Line 50-51: "CKD affects 11.2 million of population with an age-standardized rate of 8, 300 per 100,000 alone with 16,783 deaths with an age-adjusted rate of 15.4 per100,000". Perhaps if this is split into 2 sentences it would be clearer?

Line 78: "patients reduce their serum phosphorus, potassium, and protein levels..." Reduced protein levels is not desirable in dialysis patients.

Line 93: Please define "relatively validated" with regards to the questionnaire. I note appropriate steps were taken to validate the questionnaire, was it validated or not?

Lines 142-146: "Data was collected from 99 centers through various means including telephone, email, and face-to-face interviews (due to the Covid-19 pandemic). However, data from the remaining 50 centers were collected via physical interview, visiting respective centers and recruited individuals directly involved with dialysis patients in dialysis facilities such as doctors, nurses, nutritionists and dialysis assistants" What is the difference between face to face interviews and physical interviews?

Line 201: "...had excess". Access not excess, please correct.

Lines 201-204: "In addition, the number of dialysis machines per privately-owned center [20 (10-32)] was significantly higher than that of government-owned centers [6 (4-9)]. Only 6.5% of private centers had more than 20 machines, compared to 43.7% of government centers (χ2 = 24.698, p<0.001)." These sentences appear contradictory: in the first sentence it says the number of dialysis machines in privately-owned centers were significantly higher than government-owned centers. In the second sentence, it states that 6.5% of private centers had more than 20 machines compared to 43.7% of government centers... Please clarify.

Lines: 213-215: "A total of 103 DFs (69.2%) reused a dialyzer several times, while only 46 DFs (30.9%) used a dialyzer for one time, with private centers (41.7%) being more common" It may improve clarity if the authors refer to Table 1 for the dialyzer re-use patterns.

Why the differentiation between private and NGO. Are the NGOs, non-profit while the private centers are for profit?

Line 243: "It appears there is no registered dietitian (RD)as conventionally understood [29]" Please define the difference between a dietician and a nutritionist. The reference given here was a position statement on nutrition and athletic performance by the dieticians of Canada and the American college of sports medicine. The journal it was published in was not mentioned.

Lines 278-279: "The same scenario occurred regarding lack of association of ‘sector distribution of DF’ with nutrition education provision (p > 0.05) which was found available in 102 DF (68.5%)." Please clarify this sentence.

Regarding Table 1, please clarify the following:

# of dialysis machines: Total: 7 (5-11) Government: 20 (10-32) Private: 6 (4-9) NGO: 8 (7-11)

# of patients/center Total: 30 (15-56) Government: 68 (40-165) Private: 25 (15-47) NGO:35 (16-55)

The numbers do not add up to the total in each of these columns.

Also in table I: Category of patients, the listed categories do not concur with the number of patients/center listed above.

References are not well written. A lot of the references have journal names missing examples, references 1,2,4,5,6,8,9 to mention a few. References 9 -10 are the same.

Some of the references are not relevant to the text for example: Reference 5 from lines 53-55: "Although the exact prevalence of CKD in Bangladesh is unknown, Anand et al.[4] estimated that it affects 16% to 18% of the urban population, with 11% of these being in stages III to V [5]". Reference 5 here, refers to the US renal data system 2017.

In your discussion: lines 401-407: "To enhance patient care in CKD management, it is essential to develop and certify nutritionists with advanced renal dietetic skills, particularly for managing low protein diet in CKD 3 to 4 patients. A study in Bangladesh showed that patients over 40 years with insufficient protein intake were at a higher risk of malnutrition than younger patients with adequate protein intake." This study conducted by the authors was on dialysis patients, the discussion should be relevant to this population. Nutritional needs for patients with CKD stage 3 and 4 vary from those in stage 5 who are on renal replacement therapy. "Patients over the age of 40 years" is quite vague, are the authors referring to dialysis patients over the age of 40 or the general population?

Other minor corrections: In table 3: How frequently is nutrition educated provided? Regular basis (not basic).

In summary, the manuscript should be revised by the authors, noting the comments above and proof-reading for mistakes which have not been mentioned above. References should be written using PLOS one reference guidelines.

The discussion should focus on the dialysis population.

It does add to the body of knowledge on nutritional care in dialysis patients in Bangladesh and probably other low income countries.

6. PLOS authors have the option to publish the peer review history of their article (what does this mean?). If published, this will include your full peer review and any attached files.

Reviewer #1: No

Reviewer #2: No

Reviewer #3: **Yes: **Aikpokpo Ngozi

---

## [Author Response · Author response to Decision Letter 0]

5 Aug 2023

Dear Editors and Reviewers, 

The authors would like to thank the Editor and the reviewers for their valuable comments and insights that helped to improve the manuscript. We have tried to answer all the issues mentioned by the reviewers and update some parts based on the suggestions received.

Response to Reviewer 1

1. Thanks to the authors for modifying the manuscript, however, I suggest performing the following based on my previous comments: Add a paragraph in the methods to illustrate the reasons you divided the dialysis facilities into 3 sectors. Thank you for your comments.

Answer: A paragraph is added in methods; lines (100-107).

Response to Reviewer 2

1. This manuscript is a situational assessment of Bangladesh's dialysis capacity and nutrition care. The report describes the number and the location of dialysis centres in Bangladesh, underlying that the distribution is not homogenous in the country and the number of centres is inferior compared with other countries in the world. The nutrition education of patients throughout the country is deficient and should be improved. 

Answer: Thank you for your comments.

2. The manuscript is a descriptive epidemiological report and a limited value since there is no description of the biological parameters of the dialytic procedures.

Answer: We do acknowledge the limitations of a descriptive epidemiological report that lacks detailed information on the biological parameters of dialytic procedures. Given the lack of resources for conducting a more comprehensive description of the biological parameters of dialytic procedure, our study reports only observations regarding the dialysis adequacy, knowledge and service on nutritional care in dialysis facilities in Bangladesh but this is a first step in reporting CKD care status in Bangladesh which is scarcely reported. Our situational analysis on dialysis capacity and nutrition care will hopefully lead to more comprehensive assessment of dialysis delivery. The information from our study will guide the development of targeted interventions, policy changes, and resource allocation to improve the dialysis capacity and nutrition care infrastructure in Bangladesh. 

We note similar kinds of study-

1. The state of nutrition care in outpatient hemodialysis settings in Malaysia: a nationwide survey. DOI: https://doi.org/10.1186/s12913-018-3702-9

2. Implementation and Practical Application of the Nutrition Care Process in the Dialysis Unit. DOI: https://doi.org/10.1053/j.jrn.2012.01.025

3. Adequacy of Dialysis Clinic Staffing and Quality of Care: A Review of Evidence and Areas of Needed Research. DOI: https://doi.org/10.1053/j.ajkd.2011.03.027

4. Dialysis and nutrition practices in Korean hemodialysis centers. DOI: https://doi.org/10.1053/jren.2002.29534

3. Moreover, the problem of nutrition is superficially described without specific details. 

Answer: We have taken into account the feedback provided, and in our study, we have addressed the problem of nutrition in dialysis patients with more specific and detailed descriptions. We have delved into the intricacies of nutritional challenges faced by these patients, considering overall dietary management. By providing specific details, we aim to enhance the understanding of the nutrition-related issues in the dialysis population and contribute to a more comprehensive analysis of their nutritional needs. See the lines 412-422.

Response to Reviewer 3

1. The article aims to provide a "situational report" on dialysis capacity and nutrition care in Bangladesh. While the study does address these objectives, there are various errors in the write-up which need to be corrected before publication. 

Answer: Thank you for your comments.

2. Several grammatical errors and unclear sentences. Outlined are a few: 

Line 50-51: "CKD affects 11.2 million of population with an age-standardized rate of 8, 300 per 100,000 alone with 16,783 deaths with an age-adjusted rate of 15.4 per100, 000". Perhaps if this is split into 2 sentences it would be clearer? 

Answer: Modified accordingly. (Lines 49-52)

3. Line 78: "patients reduce their serum phosphorus, potassium, and protein levels..." Reduceing protein levels are not desirable in dialysis patients. 

Answer: We take note of this comment. Modified accordingly. …..helped renal patients to reduce their serum phosphorus, potassium, and control adequate protein levels. (Lines 80-81)

4. Line 93: Please define "relatively validated" with regards to the questionnaire. I note appropriate steps were taken to validate the questionnaire, was it validated or not?

Answer: Yes, it was validated. We described the validation procedure in lines 129- 139. The questionnaire has undergone some level of validation procedures, such as face and content validation and pilot study to ensure its effectiveness and accuracy in measuring the intended constructs. However, the term "relatively validated" replaced by the word “the validated” in line 95.

5. Lines 142-146: "Data was collected from 99 centers through various means including telephone, email, and face-to-face interviews (due to the Covid-19 pandemic). However, data from the remaining 50 centers were collected via physical interview, visiting respective centers and recruited individuals directly involved with dialysis patients in dialysis facilities such as doctors, nurses, nutritionists and dialysis assistants" What is the difference between face to face interviews and physical interviews?

Answer: Due to the restrictions imposed by the COVID-19 pandemic, face-to-face interviews were being conducted outside of dialysis centers, with precautions taken to minimize the risk of infection. Conversely, physical interviews involved visiting specific dialysis centers once the COVID situation improved, allowing for interviews to take place where the interviewee is located. Lines 144-146.

6. Line 201: "...had excess". Access not excess, please correct. 

Answer: Corrected accordingly. Line: 208

7. Lines 201-204: "In addition, the number of dialysis machines per privately-owned center [20 (10-32)] was significantly higher than that of government-owned centers [6 (4-9)]. Only 6.5% of private centers had more than 20 machines, compared to 43.7% of government centers (χ2 = 24.698, p<0.001)." These sentences appear contradictory: in the first sentence it says the number of dialysis machines in privately-owned centers were significantly higher than government-owned centers. In the second sentence, it states that 6.5% of private centers had more than 20 machines compared to 43.7% of government centers... Please clarify.

Answer: Thank you for this important comment. We have corrected accordingly….. Government-owned center [20 (10-32)] was significantly higher than that of privately-owned centers [6 (4-9)]. Line: 209 & 210

8. Lines: 213-215: "A total of 103 DFs (69.2%) reused a dialyzer several times, while only 46 DFs (30.9%) used a dialyzer for one time, with private centers (41.7%) being more common" It may improve clarity if the authors refer to Table 1 for the dialyzer re-use patterns.

Answer: Modified accordingly. Lines: (220-223)

9. Why the differentiation between private and NGO. Are the NGOs, non-profit while the private centers are for profit? 

Answer: Yes, private facilities prioritize profitability, while non-governmental organizations (NGOs) focus on social or humanitarian missions. But this was not the only criteria to divide NGOs and private separately. It was observed that there are significant differences in the management of dialysis services across different dialysis facilities in terms of cost, service, nutrition care, and provision of food. A brief description is now given in methods; lines (100-107).

10. Line 243: "It appears there is no registered dietitian (RD) as conventionally understood [29]" Please define the difference between a dietician and a nutritionist. The reference given here was a position statement on nutrition and athletic performance by the dieticians of Canada and the American college of sports medicine. The journal it was published in was not mentioned.

Answer: Thank you for this important comment. We have updated the reference (now the reference number is 28). Line 252.

For the difference between a dietician and a nutritionist suggested below the link- https://www.gmit.ie/sites/default/files/media/legacy/difference-between-dietitians-nutritionists-and-nutritional-therapists-katie-garvey.pdf

11. Lines 278-279: "The same scenario occurred regarding lack of association of ‘sector distribution of DF’ with nutrition education provision (p > 0.05) which was found available in 102 DF (68.5%)." Please clarify this sentence.

Answer: Modified accordingly. There was no significant association found (p > 0.05) between the sector distribution of DF and the provision of nutrition education and it was observed that nutrition education was available in 102 DF (68.5%). Lines: (287-289)

12. Regarding Table 1, please clarify the following:

# of dialysis machines: Total: 7 (5-11) Government: 20 (10-32) Private: 6 (4-9) NGO: 8 (7-11)

# of patients/center Total: 30 (15-56) Government: 68 (40-165) Private: 25 (15-47) NGO:35 (16-55)

The numbers do not add up to the total in each of these columns. Also in table 1: Category of patients, the listed categories do not concur with the number of patients/center listed above.

Answer: As previously described in ‘Statistical analysis’ section, data were presented as median with IQR for continuous variable and Categorical variables were presented as frequency with percentage. However, we modified accordingly and a table footnote was added as “Data is presented as either n (%) or median with interquartile range (IQR)” Lines: 238

In table 1, “# of patients/center” value were presented as median with interquartile range (IQR) and “Category (# of patients)” presented n (%).

13. References are not well written. A lot of the references have journal names missing examples, references 1,2,4,5,6,8,9 to mention a few. References 9 -10 are the same.

Answer: Modified according to journal requirements and reference 10 replaced by new references [9, 10]. Line 62.

14. Some of the references are not relevant to the text for example: Reference 5 from lines 53-55: "Although the exact prevalence of CKD in Bangladesh is unknown, Anand et al.[4] estimated that it affects 16% to 18% of the urban population, with 11% of these being in stages III to V [5]". Reference 5 here, refers to the US renal data system 2017. 

Answer: Modified accordingly and an updated data on CKD prevalence of Bangladesh is added. Lines: (54-56)

15. In your discussion: lines 401-407: "To enhance patient care in CKD management, it is essential to develop and certify nutritionists with advanced renal dietetic skills, particularly for managing low protein diet in CKD 3 to 4 patients. A study in Bangladesh showed that patients over 40 years with insufficient protein intake were at a higher risk of malnutrition than younger patients with adequate protein intake." This study conducted by the authors was on dialysis patients, the discussion should be relevant to this population. Nutritional needs for patients with CKD stage 3 and 4 vary from those in stage 5 who are on renal replacement therapy. "Patients over the age of 40 years" is quite vague, are the authors referring to dialysis patients over the age of 40 or the general population? 

Answer: We agree with you completely and therefore, discussion was modified accordingly as per the dialysis population and added relevant references. Please see the lines 412-422. The authors referred to the dialysis patients over the age of 40. Lines: 423

16. Other minor corrections: In table 3: How frequently is nutrition educated provided? Regular basis (not basic). 

Answer: Corrected accordingly.

17. In summary, the manuscript should be revised by the authors, noting the comments above and proof-reading for mistakes which have not been mentioned above. 

References should be written using PLOS one reference guidelines.

The discussion should focus on the dialysis population. 

Answer: According to the suggestions provided, we have revised the manuscript and taken the following actions:

1. We have carefully reviewed the comments mentioned above and made the necessary changes to address them.

2. The manuscript has undergone a thorough proofreading process to identify and correct any additional mistakes that were not specifically mentioned in the comments.

3. References have been formatted according to the PLOS One reference guidelines, ensuring consistency and adherence to the required citation style.

4. In the discussion section, we have placed a specific focus on the dialysis population, highlighting its importance and relevance to the study.

18. It does add to the body of knowledge on nutritional care in dialysis patients in Bangladesh and probably other low income countries. 

Answer: Thank you for this compliment.

---

## [Decision Letter · Decision Letter 1]

6 Sep 2023

Dialysis capacity and nutrition care across Bangladesh: A situational assessment

PONE-D-23-11624R1

Dear Dr. Rahman,

We’re pleased to inform you that your manuscript has been judged scientifically suitable for publication and will be formally accepted for publication once it meets all outstanding technical requirements.

Kind regards,

Ankur Shah

Academic Editor

PLOS ONE

Reviewers' comments:

Reviewer's Responses to Questions

**Comments to the Author**

1. If the authors have adequately addressed your comments raised in a previous round of review and you feel that this manuscript is now acceptable for publication, you may indicate that here to bypass the “Comments to the Author” section, enter your conflict of interest statement in the “Confidential to Editor” section, and submit your "Accept" recommendation.

Reviewer #1: All comments have been addressed

Reviewer #3: All comments have been addressed

2. Is the manuscript technically sound, and do the data support the conclusions?

Reviewer #1: Yes

Reviewer #3: Yes

3. Has the statistical analysis been performed appropriately and rigorously? 

Reviewer #1: Yes

Reviewer #3: Yes

4. Have the authors made all data underlying the findings in their manuscript fully available?

Reviewer #1: Yes

Reviewer #3: Yes

5. Is the manuscript presented in an intelligible fashion and written in standard English?

Reviewer #1: Yes

Reviewer #3: Yes

6. Review Comments to the Author

Reviewer #1: (No Response)

Reviewer #3: Thank you for implementing the suggestions to this revised document. The document is much clearer and easier to read.

There are a few minor typos which I have noted in the text which I have attached.

7. PLOS authors have the option to publish the peer review history of their article (what does this mean?). If published, this will include your full peer review and any attached files.

Reviewer #1: No

Reviewer #3: **Yes: **Ngozi Virginia Aikpokpo

---

## [Editor Report · Acceptance letter]

13 Sep 2023

PONE-D-23-11624R1 

Dialysis capacity and nutrition care across Bangladesh: A situational assessment 

Dear Dr. Rahman:

I'm pleased to inform you that your manuscript has been deemed suitable for publication in PLOS ONE. Congratulations! Your manuscript is now with our production department. 

Kind regards, 

on behalf of

Dr. Ankur Shah 

Academic Editor

PLOS ONE